# Development and Validation of a Bioinformatic Workflow for the Rapid Detection of Viruses in Biosecurity

**DOI:** 10.3390/v14102163

**Published:** 2022-09-30

**Authors:** David W. Waite, Lia Liefting, Catia Delmiglio, Anastasia Chernyavtseva, Hye Jeong Ha, Jeremy R. Thompson

**Affiliations:** 1Plant Health and Environment Laboratory, Ministry for Primary Industries, P.O. Box 2095, Auckland 1140, New Zealand; 2Animal Health Laboratory, Ministry for Primary Industries, Upper Hutt 5018, New Zealand

**Keywords:** biosecurity, virology, bioinformatics, high-throughput sequencing, Oxford Nanopore, Illumina, DNA, RNA

## Abstract

The field of biosecurity has greatly benefited from the widespread adoption of high-throughput sequencing technologies, for its ability to deeply query plant and animal samples for pathogens for which no tests exist. However, the bioinformatics analysis tools designed for rapid analysis of these sequencing datasets are not developed with this application in mind, limiting the ability of diagnosticians to standardise their workflows using published tool kits. We sought to assess previously published bioinformatic tools for their ability to identify plant- and animal-infecting viruses while distinguishing from the host genetic material. We discovered that many of the current generation of virus-detection pipelines are not adequate for this task, being outperformed by more generic classification tools. We created synthetic MinION and HiSeq libraries simulating plant and animal infections of economically important viruses and assessed a series of tools for their suitability for rapid and accurate detection of infection, and further tested the top performing tools against the VIROMOCK Challenge dataset to ensure that our findings were reproducible when compared with international standards. Our work demonstrated that several methods provide sensitive and specific detection of agriculturally important viruses in a timely manner and provides a key piece of ground truthing for method development in this space.

## 1. Introduction

The New Zealand (NZ) export market is dominated by high quality agricultural products, with six of the ten top annual exports in 2021 coming from primary industries [1]. Over NZD 47 billion worth of exports, from a total of NZD 63.3 billion, in 2021 came from primary production after a significant growth in many of the major export commodities such as dairy products, meat, and wood [1,2]. In 2021 (the year to 30 June 2022) the NZ dairy sector generated NZD 19.1 billion in export revenue, roughly equal to that revenue of meat and wool ($10.4 billion NZD) and horticulture (NZD 6.6 billion) exports. NZ is currently free from many of the plant and animal viruses which compromise production and export value. In a country whose economy is so tightly linked to primary production, employing effective biosecurity measures to maintain this state is greatly desired amongst growers and exporters [3]. Safeguarding these industries is therefore a top priority for the Ministry for Primary Industries, to maintain and expand the productivity of NZ.

The introduction of exotic pathogens is a constant concern for NZ as international trade, while such an integral part of the economy, is also a major pathway for the introduction of economically significant viruses [4]. Incursion and subsequent establishment of regulated viruses would result in a decline of yield and quality of domestically grown, high-value products. Beyond the obvious reduction in marketable value for lost or disfigured crops and animals, invasions carry implications for trade agreements, as preferential access to international markets is often contingent upon the absence of specific pathogens. For example, an outbreak of Foot-and-mouth disease (FMD) could freeze NZ meat exports for months, resulting in billions of dollars of lost export value [5]. Similar consequences may occur with the introduction of the viruses responsible for African swine fever and Lumpy skin disease—losses of production and export value, as well as eradication, compensation, and surveillance costs may have considerable economic impact on primary industries [6,7]. Although these viruses are currently absent from NZ there are increasing concerns for their introduction into Australia, a major trading partner of NZ [8,9,10,11]. To address the ever-increasing risk of virus incursions, high-throughput sequencing (HTS) is now becoming routine within the NZ biosecurity sector, particularly when performing plant virus diagnostics [12]. The benefits of HTS as a diagnostic tool are widely acknowledged [13,14,15,16,17,18] and this approach has gained favour within NZ primarily for its broad target range when compared with traditional diagnostic techniques.

The use of HTS platforms for the sequencing of shotgun metagenomic (and metatranscriptomic) libraries is a technique commonly applied to the study of microorganisms [19,20], particularly in instances where cultivation of microorganisms is challenging. The applicability of high-throughput shotgun sequencing to biosecurity has long been recognised [13] and there are now numerous examples of this technology being used in surveillance and diagnostic settings [15,21,22,23,24,25,26,27]. Unlike traditional diagnostic techniques, such as ELISA and PCR, there is currently little standardisation of HTS laboratory methods and bioinformatic analyses when they are applied to pathogen diagnostics. This is particularly true when an HTS approach is used in the absence of prior knowledge of virus identity and simply mapping reads to a reference genome is not appropriate. For example, of the exemplar references above, each of those published since 2018 performed significantly different techniques for classifying sequence reads and obtaining a final diagnosis.

The variety of analysis methods and classification databases provides flexibility when performing analyses, allowing diagnosticians to customise their approach to specific pathogens of interest. However, this lack of standardisation results in a fragmentation of practices and workflows between institutions and countries, despite efforts to create harmonized approaches [28]. While efforts have been made to develop tools to automate the detection of viral sequences from HTS libraries, such tools often perform poorly when applied beyond their intended use. In a recent literature review, Nooij and colleagues [29] compared performance reports for 25 virus detection workflows and reported an average sensitivity of just 0.67 across all tools and datasets evaluated. Superior results are reported when general-purpose metagenome classifier tools have been applied to virus detection in clinical samples [30,31,32]. Owing to such observations, when performing virus discovery in plant HTS libraries there remains an overwhelming bias towards the use of virus-agnostic analysis pipelines. For example, from 71 HTS-based virus detections summarised by Ibaba and Gubba [33], 65 performed their main detection of viral sequences using short-read assembly, BLASTn or BLASTx queries to the NCBI GenBank database, and screening for viral proteins using Kaiju [34]. Similarly, a 2017 review of HTS applications in plant diagnostics highlighted seven instances of virus identification being performed by HTS, all of which relied on a manuscript-specific BLAST-based approach rather than a published virus pipeline [35]. Consequently, HTS-based initiatives focusing on virus discovery still favour alignment-based search tools or protein fold models for performing initial detections [36,37,38].

The discrepancy between the performance of custom-built virus detection tools and the more generic forms of sequence annotation may be explained by the observation that sequence databases are dominated by particular lineages, which likely results in bias. Many recently developed tools are not specifically calibrated on pathogens or risk organisms [39,40,41,42,43,44] and the datasets used in training are often rich in phage sequence [45,46,47,48,49] which contribute towards global sensitivity and specificity measures without being relevant to diagnoses in eukaryotes. We have previously reported the development of an in-house laboratory protocol for sequencing viruses using the Oxford Nanopore Technologies (ONT) MinION device, which allows us to generate cDNA libraries from infected plant material without the need for polyadenylated RNA capture [12]. However, with uncertainty surrounding the efficacy of existing high-throughput analysis pipelines, the subsequent bioinformatic workflow relied on a BLASTn query against the full NCBI nucleotide database. While undoubtedly thorough, this approach requires significant computational resources, can potentially require days of computation time to generate initial findings, and is not appropriate for samples sequenced using short read technologies without some form of genomic assembly to reduce the number of query sequences to be examined.

As biosecurity diagnoses are performed within tight reporting timeframes, the long classification times, coupled with the inherent uncertainty around the run time of any specific BLAST job led us to explore alternative methods for generating sensitive and accurate classifications with more timely and predictable results. We sought to identify tools which could perform tasks such as short-read assembly and prediction of protein coding regions quickly and robustly, so that an end-to-end diagnostic pipeline could be constructed to reliably screen sequence libraries from a range of hosts to provide timely indications of virus presence in samples. We identified a variety of existing tools and classification databases with the express purpose of detecting virus sequences in HTS libraries while assessing both their ability to identify plant and vertebrate viruses, and their specificity against host or other sequence contaminants. We constructed candidate analysis workflows from the best-performing tools and then tested said workflows against 165 synthetic libraries. Each synthetic library consisted of a plant or animal sample spiked with a virus of interest to the NZ agricultural or horticultural sector, and sequences were simulated for both Oxford Nanopore and Illumina sequencing platforms. In addition, we applied our methods to 18 samples available through the VIROMOCK Challenge [50] a recognised international platform to benchmark bioinformatic pipelines for virus detection.

## 2. Materials and Methods

### 2.1. Data Collection and Creation of Test Data

Viral genomes were downloaded from the NCBI RefSeq and GenBank database on the 9th of April 2020. Downloads from RefSeq were performed by iterating through the contents of the ‘ftp.ncbi.nih.gov://genomes/refseq/viral/’ and downloading each species entry via rsync. For each species record downloaded, genome sequence files and GenBank files were preferentially selected from the ‘latest_assembly_versions/’ folder if present, or the ‘all_assembly_versions/’ folder if the first option was empty. Genome records marked as ‘suppressed’ were ignored along with any GenBank records which had a corresponding RefSeq version of the assembly. Viral species were matched to a broad host lineage using the information available at the NCBI assembly reports. In the case of 3709 entries a host record was not provided but could be inferred from the virus name. These genomes are collectively referred to as the ‘target’ dataset.

To test the specificity of databases and analysis pipelines, genomes from the fungus, invertebrate, plant, protozoan, vertebrate_mammalian, and vertebrate_other locations in the NCBI FTP server were downloaded and parsed in the same manner. The sequence data for each lineage was additionally screened to remove identical contigs between records. A dereplicated set of prokaryotic (bacterial and archaeal) genomes were obtained from the Genome Taxonomy Database (release 95) [51,52]. These genomes are collectively referred to as the ‘non-target’ dataset. Protein sequences obtained from the non-target dataset were further dereplicated by clustering of protein sequences using the ‘linclust’ function of MMSeqs2 (version 12.113e3) [53]. Clustering was performed using 80% sequence similarity for the prokaryote lineage, and 95% sequence similarity for the remaining non-target datasets, to reduce the number of queries submitted for profiling tools.

### 2.2. Creation of Custom Viral Sequence Databases

To achieve faster classification rates than that achieved with BLAST queries against the NCBI nt and nr databases, we constructed virus-specific databases from the viral genomes and gene predictions obtained when constructing the target dataset. As there are many viral sequences for which a full genome are not yet available additional sequences were obtained by extracting all viral sequences from the NCBI nt and nr databases (July 2021 release) then dereplicating the results to a more manageable size. For nucleotide data, the nt sequences were grouped according to NCBI species name and clustering with vsearch. We initially performed grouping by NCBI species name and clustered intra-species sequences at 90% sequence identity, then also created datasets by truncating species names at the key words ‘virus’, ’viroid’, and ’phage’ to produce broader groupings, followed by intra-species sequence clustering with 90%, 97%, and 99% sequence identity thresholds.

When performing nucleotide clustering, two lineages were removed from the input data as they comprised a disproportionately large percentage of reads in the raw data. The first of these corresponded to the SARS-CoV-2 lineage from which genomes have been sequenced at a disproportionately high rate in the last two years. In this instance, we inserted a single representative sequence (NC_045512.2) to represent this lineage. Clustering was also impeded by a large grouping of unclassified viruses belonging to the family *Myoviridae*. In this instance, the cluster was removed from the database as; (a) viruses of the family *Myoviridae* infect bacteria and are therefore unlikely to be relevant to the aims of this work, and (b) there were many named isolates of this virus in the database, so the loss of this particular lineage was not deemed detrimental to the taxonomic coverage of the clustered sequence set. Protein clustering was performed by filtering protein sequences fewer than 25 amino acid residues in length, then grouping by either species name or truncated species name grouping and clustering proteins at 90% sequence identity using usearch (11.0.667) [54].

### 2.3. Creation of Custom Viral HMM Databases

There are numerous high-quality HMM databases of viral protein clusters currently available, however, many are rarely updated and several of the more recent databases comprise tens of thousands of models, making annotation a slow process. We sought to improve upon these resources by producing a database which could compete in terms of specificity and sensitivity while minimising the time required to classify proteins. Protein sequences were extracted from GenBank record files downloaded for the target dataset and identical proteins were removed using seqmagick. The dereplicated proteins were then purged of polyprotein sequences using an all-against-all BLASTP search with DIAMOND (version 0.9.32) [55]. Results were filtered to remove self-hits, and polyproteins were identified using the working definition of the vFAM database project, namely that a polyprotein is one in which greater than 70% of the protein sequence is covered by two or more non-overlapping shorter protein sequences. From an input set of 686,558 unique protein sequences, 33,562 were identified as polyproteins by this workflow and removed from the dataset.

Protein clusters were produced using multiple methods, based on either static of dynamic cluster definitions. Clustering was initially performed with usearch, clustering proteins into groups using identity thresholds of 50%, 60%, 70%, 80%, and 90%. A parallel clustering effort was performed using mcl (version 14.137) [56] to cluster the results of an all-against-all DIAMOND BLASTp query. Inflation values of 1.2, 1.4, 2.0, 4.0, and 6.0 were used to cluster the output using the ‘--stream-neg-log10′ parameter and transformation function ‘-stream-tf ‘ceil(200)’’ to convert e-values into edge weights for the input graph. Inspection of the outputs using the ‘clm dist’ and ‘clm info’ tools was performed and from the initial clustering results, a value of 4.0 was selected to define protein clusters. A hybrid clustering method was also applied, in which the proteins not successfully clustered by usearch at each identity threshold were then clustered by mcl. The resulting protein clusters were appended to the original usearch result as supplemental protein clusters.

To reduce the number of candidate protein clusters required to produce a comprehensive sampling of diagnostically relevant viral proteins, the output of each clustering iteration was evaluated. Protein clusters were scored in terms of their coverage of viral families and subfamilies using a modified scoring metric developed by Bobay and Ochman [57]. For every (sub)family grouping, each protein cluster was scored according to how many members of the grouping had contributed a protein into the cluster. If a cluster covered a sufficient portion of any (sub)family grouping it was retained as a sufficiently conserved for diagnostic purposes. The threshold for retention was set as:$$min\left(\begin{array}{c} 0.85\\ 1-\frac{1}{n} \end{array}\right)$$

Thus, for large groupings the coverage requirement was 85%. For groupings with fewer than 7 genomes, it is not possible to score greater than 85% unless all members contribute to the cluster, so the threshold was set to retain clusters which were not represented by a single genome in a grouping. A multiple sequence alignment was performed on each remaining cluster using a range of tools, as there is no single alignment tool which consistently outperforms all others [58]. MAFFT (version 7.310) [59,60], MAFFT-L-INS-I [61], Clustal-Omega (version 1.2.4) [62], ClustalW2 (version 2.1) [63], and MUSCLE (version 3.8.31) [64] were applied to each cluster and the alignment which introduced the fewest gaps was selected as the representative alignment. Each sequence alignment was then converted to an HMM profile using hmmbuild (version 3.2.1) [65,66].

The target and non-target protein sequences were classified against each database using hmmsearch. For comparative purposes, we downloaded and tested the following publicly available databases using the same datasets; pVOGs [67] (June 2017 release), VOGDB (https://vogdb.org/, release 99), RVDB (version 18.0) [68], vFAM-A and vFAM-B [69] (February 2014 release), and EggNOG [70] (viral OGs, October 2015 release). We also downloaded the classification databases used in viral detection tools IMG/VR (version 3.0) [71], CheckV, (version 0.6) [72], and VirTool (https://www.virtool.ca/, November 2017 release). Hits were filtered to retain matches with a sequence-level e-value of ≤0.001 for all databases except for CheckV, for which we applied the cluster-specific bitscore thresholds specified in the documentation.

### 2.4. Testing of Sequence-Based Classification Tools

The dereplicated nucleotide and protein databases were tested for sensitivity and specificity. We defined sensitivity as the proportion of viral sequences which were correctly identified as such, and specificity as the proportion of the nontarget sequences correctly rejected by the classifier following appropriate filtering for each nontarget lineage (Table 1). For alignment-based, nucleotide/nucleotide classification we tested BLASTn (version 2.10.0) [73,74], usearch, vsearch (version 2.15.1) [75], and MMSeqs2 (version 12.113e3). Nucleotide/protein (translated) searches were performed using DIAMOND and MMSeqs2, and protein/protein searching was performed using BLASTp, DIAMOND, MMSeqs2, and usearch with both the usearch_global and ublast algorithms. Nucleotide/nucleotide and protein/protein searches required 70% identity over 50% of the query length and an e-value of less than 0.001 to be accepted as valid matches. Translated searches were performed using the same identity and e-value requirements, but without a minimum coverage value to account for instances in which open reading frames may be identified within a subset of a nucleotide sequence (i.e., where a read spans the start or stop boundary of the transcript). For nucleotide/nucleotide searches, query contigs were broken on poly-N scaffolding regions to prevent the interference of artificially induced ambiguous regions on alignment scores.

Nucleotide classification via k-mer frequency profiles was performed using Kraken2 (version 2.1.1) [76], KrakenUniq (version 0.5.7) [77], CLARK (version 1.2.6.1) [78], MetaCache (version 1.1.1) [79], and ganon (version 0.3.4) [80]. For Kraken2 the pre-formatted k2_standard_20201202 and k2_viral_20201202 databases were downloaded for classification. For all other tools, the tool-specific database creation commands were used to build the databases. Downloads for database construction commenced on the 23rd of December 2020, corresponding to release 203 of GenBank.

Pipelines and software suites developed for the detection of viral sequences in HTS libraries were identified through a literature review and tested against the target dataset. The tools installed were as follows: MiCOP (unversioned) [41], FastViromeExplorer (unversioned) [40], ViromeScan (not versioned) [39], VirFinder (version 1.1) [48], VirSorter (version 1.0.6) [46], VirSorter2 (version 2.1) [47], VirTool (https://www.virtool.ca/, version 3.9.6), and DeepVirFinder (version 1.0) [49]. All software was downloaded from the respective sources in January 2021, so for tools which do not publish with version information the current release at that time point is the version used in this analysis.

Each tool was used to classify the target and non-target datasets using their default detection thresholds. When no score cut-off was specified, the highest ranked hit per query sequence was retained as the result of classification. Multiple testing correction was not performed DeepVirFinder predictions, as the target dataset is exclusively viral sequences and correcting for multiple testing would suppress legitimate identifications. For nucleotide analysis with VirTool, mapping was performed using bwa (version 0.7.17) with the ‘-x intractg’ flag because long read mapping is not recommended by bowtie2, the tool used internally by the VirTool workflow.

### 2.5. Construction of Mock Libraries for Workflow Evaluation

A selection of plant and vertebrate viruses deemed of high importance to the NZ agricultural and horticultural sectors were identified. A total of 156 viruses comprising both RNA and DNA genomes were selected, and genomes from 16 host species were downloaded to simulate libraries of infected host material (Appendix A). To model the skew of transcript abundance observed in RNA sequencing data, a negative binomial distribution of transcript abundance was produced by mapping ONT sequence data obtained during diagnostic sequencing against the NCBI tomato transcript reference (version GCF_000188115.4). The reference transcriptome was clustered using CD-HIT-EST (version 4.8.1) [81] with the parameters ‘-c 0.95 -n 10 -d 0 -T 10’ then ONT sequencing data produced from a previously published library preparation technique [12] were then mapped to the transcript reference using minimap2 (version 2.18) [82] using the ‘map-ont’ method. A negative binomial distribution of transcript coverage was estimated using maximum likelihood estimate of parameters (n = 0.052, *p* = 0.011) with the SciPy statistical package [83]. To create simulated libraries, the transcriptome reference sequences were obtained for each host species which were similarly dereplicated using CD-HIT-EST. Five randomly sampled transcript abundance libraries were created using the negative binomial model, and randomly assigned to each virus of that host.

Mock sequencing libraries were constructed to mimic the results of the ONT MinION sequencing platform using BadRead (version 0.2.0) [84] with the ‘nanopore2020′ error and q-score models. For DNA viruses and host genomes, libraries were simulated using default parameters. For RNA viruses and transcriptomes, one of the five transcript abundance libraries were sampled at random and BadReads were run using the parameters ‘--chimeras 0 --length 250,180′. Length parameters correspond to the mean and standard deviation of sequence lengths in the tomato sample used in mapping above. All libraries were simulated to produce at least 1 million host reads and 10,000 virus reads. Illumina HiSeq library simulation was performed using the same reference data as for ONT, using InSilicoSeq (version 1.5.2) [85] with the default HiSeq error model to produce 100 million paired end reads per host and 50,000 per virus.

For each virus/host pairing, the ONT library size was set as 400,000 sequence reads, approximately equivalent to the result of a 24 h ONT Flongle run [12]. For each ONT library the composition of viral reads was randomly sampled as an absolute value from a normal distribution with a mean of 0.06 and standard deviation of 1.1. These parameters were obtained by fitting a distribution to the percentage of viral reads identified in virus-positive plant material diagnosed at the Plant Health and Environment Laboratory during 2021 with the method described by Liefting et al. (2021). For HiSeq libraries, a sequencing depth of 40 million paired-end reads was selected. The same distribution was used to generate viral abundances, but values were reduced by a factor of 10 to simulate a lower viral titre. As an additional means of comparison, 18 datasets from the publicly available VIROMOCK challenge (https://gitlab.com/ilvo/VIROMOCKchallenge, accessed 05/08/2021) were downloaded and tested to assess the robustness of the workflows.

### 2.6. Application of a Diagnostic Workflows to Mock Libraries

The top performing classification tools were selected from the results of sensitivity and specificity testing based upon these two factors, and their average run duration. This selection of tools comprised BLASTn for nucleotide alignment search, DIAMOND (BLASTx) for translated nucleotide alignment search, DIAMOND (BLASTp) and usearch (usearch_global) for protein alignment search, and Kraken2 for k-mer frequency profiling. None of the virus detection pipelines performed sufficiently well to be considered for further testing (see Results below). To simulate a complete diagnostic event, a series of analysis options for each library were arranged to perform the following steps upon the raw libraries: quality filtering and adapter removal (mandatory), host mapping and removal of host reads (optional), assembly (Illumina only), and protein coding sequence prediction prior to classification of reads (Figure 1).

When working with ONT sequence data, quality filtering was performed with NanoFilt (version 2.6.0) [86] and adapter removal with Porechop (version 0.2.4) [87]. Sequences were mapped to the host genome using minimap2 (version 2.20) with the ‘map-ont’ or ‘splice’ settings for DNA and RNA libraries, respectively. For predicting protein coding regions for protein annotation, the tools prodigal (version 2.6.3) [88] and TransDecoder (https://github.com/TransDecoder, version 5.5.0) were tested. Prodigal was run using anonymous (metagenomic) mode, to avoid training sample-specific gene models. TransDecoder was applied in two ways, using the basic pairing of the TransDecoder.LongOrfs and TransDecoder.Predict functions (hereafter referred to as TransDecoder Base) and with the use of homology searches to guide ORF retention between the LongOrfs and Predict functions (TransDecoder Guided). For guided prediction, the initial set of candidate peptides identified by TransDecoder.LongOrfs were annotated using hmmsearch (v3.3) against the Pfam (version 32.0) [89] database and DIAMOND BLASTp (v2.0.6) against the SwissProt (release 2020_06) [90] database prior to prediction.

Quality filtering and adapter removal were performed for simulated HiSeq and VIROMOCK libraries using fastp (version 0.20.0) [91]. Host sequence removal was performed by mapping reads to the reference genome using bowtie2 (version 2.4.4) for DNA and HISAT2 (version 2.2.1) for RNA libraries [92,93]. Successfully mapped reads were identified and removed using samtools (version 1.12). Assembly was performed with three different assembly tools for each library type, selected for their documented strengths when assembling metagenomic datasets [94]. DNA libraries were assembled using MegaHIT (version 1.1.4) [95], and the metaSPAdes and metaviralSPAdes routines from the SPAdes assembler (version 3.15.0) [96,97,98]. For RNA libraries MegaHIT, rnaSPAdes [99], and rnaviralSPAdes [100] were applied. Gene prediction was performed using the same tools and parameters as the ONT libraries.

Each traversal from raw sequence to results (Figure 1) is hereafter referred to as an analysis iteration. Each analysis iteration was assessed in terms of each tools ability to recover the exact strain and species of the virus spiked into each library.

## 3. Results and Discussion

### 3.1. Creation of Target and Non-Target Datasets for Tool Testing

A total of 26,206 viral genomes were obtained from NCBI, consisting of 9011 from RefSeq and 17,195 from GenBank. When accounting for entries which were duplicated between repositories, a final count of 17,876 genomes from 17,231 viral species were obtained, which included 47 viroid genomes. Host information from NCBI was used to organise the viruses and viroids into broad host categories; 1992 plant, 2494 vertebrate, 1931 invertebrate, 313 fungus, 72 protozoan viruses, and 47 plant viroids. The remaining genomes were collapsed to the category of ‘other’ host consisting of viruses infecting algae, diatoms, prokaryotes, and viruses for which host information could not be established. From these viruses, 27,340 contigs were extracted and dereplicated into 27,029 unique sequences with seqmagick (https://fhcrc.github.io/seqmagick/, version 0.7.0). Protein coding sequences were extracted from the GenBank files to produce an initial dataset of 1,088,425 amino acid sequences which were similarly dereplicated to 686,559 unique sequences. Nucleotide sequences for the non-target dataset were extracted and reduced to unique contigs, and non-target protein sequences were dereplicated by clustering with the MMSeqs2 ‘linclust’ function using thresholds of 80% identity over 90% alignment coverage for prokaryotic proteins, and 95% identity for all other lineages (Table 1).

### 3.2. Creation of Custom Sequence Databases

In addition to the genomes obtained when constructing the target dataset, a total of 3,308,823 viral nucleic acid sequences were obtained from the NCBI nt database, along with 8,160,005 protein sequences from nr. These reference sets of viruses were filtered and dereplicated into candidate classification databases. The number of lineage clusters and representative sequences for each database and the working title for each candidate database can be found in Table 2.

### 3.3. Creation and Assessment of Custom Viral HMM Databases

A total of 12 candidate viral HMM databases were created from the target dataset. Following deduplication and removal of polyprotein sequences, 652,996 sequences remained for cluster generation. Clustering of these sequences was performed using usearch (50% to 90% identity thresholds, 10% increments) and mcl (inflation parameter 1.4 and 4.0). Five additional cluster sets were created by combining these techniques, applying mcl clustering (I = 4.0) to the proteins which were not clustered in each usearch iteration. Clustering methods yielded between 130,000 and 550,000 sequence clusters depending on the method used, although once singleton clusters and clusters with insufficient viral coverage (see Section 2.3) were removed from the result this number was significantly reduced (Table 3).

For each cluster which passed coverage filtering, a multiple sequence alignment was performed using four different tools, with two alignment algorithms applied using MAFFT. The alignment which introduced the fewest gaps per cluster was selected as the ‘optimal’ alignment and an HMM profile was generated from the alignment using hmmbuild. The complete set of HMMs for each clustering approach were then pooled and a complete database produced using the hmmpress. To assess the sensitivity of each method, the target dataset was classified using each candidate HMM database. As the cluster selection method deliberately discarded many clusters of viral protein, sensitivity was scored at the genome level rather than the level of protein. Specifically, we counted the number of viral genomes from which at least one protein was identified (Appendix A).

Raw sensitivity values were highest across mcl and mixed clustering methods but promising from all approaches. Four candidate databases were selected for further evaluation—usearch_50, mcl_1.4, mcl_4.0, and mixed_90. Specificity scores were assessed at the level of protein coverage, rather than genome coverage, and results were lower than the sensitivity scores across all databases (Appendix A). This finding is not surprising, as the HMM approach is a method aimed at recovering ancestral relationships which can capture protein motifs and signatures which are distantly related through evolutionary time or may have been the result of convergent evolution. This functionality is desirable in the context of our workflow, as we envision HMM searching as a secondary means of analysis to identity uncharacterised and new-to-science viral proteins rather than a means for obtaining exact matches to characterised pathogens.

Between the four candidates there was no clearly superior clustering method for producing the final classification database. Each technique produced variable results in terms of sensitivity and specificity, but not surprisingly these values were negatively correlated—clusters which were able to capture a larger variety of viral proteins produced a greater number of false positive hits amongst non-target proteins. We ultimately favoured databases with higher sensitivity as results of these searches would always be validated, whereas poor sensitivity will result in false negative results which are unlikely to be further investigated. Between the options tested both the mixed_90 and mcl_4.0 databases demonstrated the highest sensitivity for plant and vertebrate viruses (Table 4). Between these two options, average sensitivity was equivalent when e-value filtering was applied to the results and we selected the mcl_4.0 database due to its slightly higher specificity and drastically shorter run times.

### 3.4. Classification and Database Profiling

Classification of nucleotide sequence data was performed using alignment-based and k-mer frequency profiling tools, as well as a set of software tools designed specifically for identifying viral sequences in metagenomic datasets. The sensitivity of each tool was tested against the target dataset and specificity was assessed using the non-target data (Figure 2, Appendix A). For the alignment-based tools, the BLASTn tool provided the greatest sensitivity and specificity, returning near-perfect coverage of the test datasets. A strong result was also observed with the MMSeqs2 easy-search functionality. Classifications with both usearch and vsearch were unable to complete, as run times when classifying with these tools were so prohibitively long that it was not deemed feasible to utilise them in a practical setting. We speculate that the reason for these poor times is a consequence of the indexing process performed by these tools, as both were originally developed for working with amplicon sequencing datasets in which databases are generally smaller and much more homogeneous. We did observe that usearch performed extremely well when classifying protein sequence classifications, where a more heterogeneous population of target sequences would be expected. Classification of plant virus sequences using k-mer profiling tools was generally successful across all tools tested, with a median sensitivity of 0.9584. However, sensitivity towards vertebrate viruses was notably lower, ranging from 0.753 to 0.866 (Figure 2). Specificity values were more variable, but Kraken2 was a clear leader with an average specificity score of 0.9996 when using the standard database against both plant and vertebrate viruses (Figure 2, Appendix A).

For the sequence alignment and k-mer tools the time needed to complete analysis was also considered. Although time is not necessarily the most important factor when performing an analysis, it is still a consideration due to reporting windows. Run times were low for most k-mer classification tools, typically requiring less than 1 min to classify the target dataset (Appendix A). Surprisingly, BLASTn proved quicker in performing classification than MMSeqs2, requiring as little as half the time to classify the target dataset (Appendix A). Both usearch and vsearch performed so poorly under the time metric that they were discarded from the nucleotide classification section of our analysis as they would not be feasible in a real diagnostic setting.

In addition to standalone classification tools, a series of software pipelines designed for the identification of viral sequences from HTS libraries were trialled. Across the board, these tools performed poorly for sensitivity of plant and animal viruses, with only five out of 13 tool/database combinations scoring a sensitivity greater than 0.8 for either plant or vertebrate viruses (Table 5). We believe that this poor sensitivity is a consequence of the large number of phage sequences used in their development, either by design or as a consequence of NCBI database skew. For example, DeepVirFinder achieved high sensitivity for the ‘other’ group (Table 5) which included prokaryote-infecting viruses despite performing poorly in other lineages.

Protein classification was performed using both alignment-based search tools and HMM searches using hmmer. Sensitivity of all alignment-based search tools was high, with over 98% of viral proteins correctly identified as viral across all host types (Figure 3, Appendix A). Translated searches were slightly less sensitive when considering plant viruses but performed equally well as protein/protein searches for vertebrate viruses.

Based on these findings we identified a trusted set of tools and databases which provided the best trade-off between sensitivity, specificity, and computational run time. We then sought to test these tools against noisier datasets, representative of what would likely be encountered during real diagnostic settings.

### 3.5. Recovery of Nucleotide and Gene Sequences from Simulated Libraries

Mock infections were simulated for 24 DNA virus infections (20 vertebrate, 4 plant) and 141 RNA virus infections (46 vertebrate, 95 plant). The selection of viruses was chosen to include viruses which are routinely identified in the Ministry for Primary Industries virology laboratories and emerging viruses which may prove to be of economic importance in the future. For each infection, sequences representing the outcome of Oxford Nanopore Technologies MinION and Illumina HiSeq platform were simulated. Mock ONT libraries were created at a depth of 400,000 sequences, mainly from the host organisms, and HiSeq libraries contained 40 million sequences, with a viral load approximately 10% that of the ONT libraries. These values were selected to reflect our current practice of performing ONT sequencing exclusively on symptomatic plant material, whereas Illumina sequencing is favoured when virus symptoms are not always apparent.

Each mock library was analysed through a variety of analysis pathways (Figure 1) which comprised the removal of low-quality sequences, followed by optional removal of host sequences using a workflow appropriate for the library. For HiSeq libraries, assembly was performed using MegaHIT and SPAdes [94]. DNA libraries were assembled using the metaSPAdes and metaviralSPAdes routines, and RNA libraries were assembled using rnaSPAdes and rnaviralSPAdes. When working with VIROMOCK libraries, the Illumina workflow was followed without host removal, as not all samples in the dataset contain host sequence. Assembly was not performed with ONT libraries as, to our knowledge, there is currently no de novo metatranscriptome assembler available. While we have successfully assembled cDNA Oxford Nanopore datasets to produce viral genomes [12] the performance of any given tool is variable when working with such data, and this instability does not lend itself well to a hands-off approach. As the primary objective of this work is to identify methods for the identification of viral sequences, not to produce representative genomes, we opted to skip a general-purpose assembly step for the ONT libraries. For these samples, fasta sequences were produced directly from the quality-filtered sequences using seqmagick.

To perform protein sequence analysis, prediction of open reading frames in nucleic acid sequences and subsequent translation into an amino acid sequence is required. However, in a diagnostic setting the organism to be detected is not known in advance and therefore a tailored gene prediction model cannot be applied to the data. While much work has been done to address this issue in the field of environmental metagenomics and prediction of coding sequences in prokaryotic lineages, viruses exhibit features not always encountered in prokaryotes, such as non-canonical start codons, readthrough transcription, leaky-scanning, and in some cases gene splicing [101,102,103,104,105,106,107]. We therefore compared prodigal, a tool designed for *ab initio* prediction of coding sequence in metagenomic datasets, with TransDecoder, designed for prediction of coding sequences with RNA-Seq data (Figure 1).

When assessing the results of the HiSeq libraries, it was quickly apparent that not all assembly options were equally robust against variations in the input data. The SPAdes assembler occasionally failed to produce an assembly, but this was particularly noticeable in the case of the metaviral assembly routine, which did not produce a successful assembly for DNA viruses when the host sequence was removed prior to assembly (Table 6). Similarly, protein coding prediction was not equally robust across all methods as only prodigal was able to produce protein coding sequence predictions from all input libraries (Table 6). We observed that prediction of protein coding sequences by TransDecoder suffered when host sequences were removed from Illumina assemblies. We attribute this observation to the TransDecoder algorithm attempting to generate a training model from the input data [108], as it is likely that when there is insufficient sequence data to produce initial models the subsequent prediction will suffer. In instances where assembly and gene calling were successful the results were still evaluated, but the less reliable performance of these tools meant that results were primarily focused on the prodigal predictions.

### 3.6. Performance of Nucleotide-Based Classification Approaches

Following the analysis of each mock infection through each analysis pathway the results were screened to determine whether sequences from the library were correctly identified as belonging to the strain and species of the spiked virus. Classification of the nucleotide sequences was performed using BLASTn with each database described in Table 2, Kraken2 with both databases, and DIAMOND BLASTx using the protein databases tested in Table 2.

Amongst nucleotide classification methods we noted that performance across the five in-house databases was not equal. For the ONT dataset, the nt_derep4 candidate database correctly identified all DNA viruses and 138 of 141 RNA viruses at the strain level but at the species level, the nt_derep1, nt_derep2, and nt_derep3 databases all achieved 100% sensitivity across RNA virus samples. The sensitivity of the ONT method was not affected by removal of host sequences but amongst the HiSeq libraries performance was variable and clearly influenced by host removal. Despite its superior performance in terms of completing assemblies, MegaHIT performed slightly poorer than SPAdes when run in RNA mode for identifying the exact strain of virus, although for species-level identification both tools performed equally well. MegaHIT performed marginally better than SPAdes (metagenomic mode) when assessing DNA samples at the strain level but following the removal of host sequences this result was reversed and metaSPAdes was able to perform equally well or better depending on the taxonomic resolution (Figure 4). We therefore tentatively conclude that for assessing DNA viruses MegaHIT is the superior option, as is SPAdes in transcriptomic mode for RNA viruses, but the performance of both tools is extremely close, and the effect of database and classification tool plays a far greater role in determining sensitivity.

K-mer profiling classification did not perform as well in simulated data as the initial comparisons against the target dataset would suggest. Across all library combinations, Kraken2 demonstrated a median sensitivity of 81.7% for strain-level and 89.6% for species-level detections (Figure 4). These values were comparable to the classification sensitivity of the BLASTn genome_nt database (81.2% and 91.0%, respectively) but inferior to results obtained by the more comprehensive versions derived from the NCBI nt database. This lower sensitivity may be a consequence of the Kraken2 and genome_nt databases only including representative genome sequences, as new and emerging sequences are typically first detected as amplicons and partial sequences.

Translated protein queries, performed via the DIAMOND BLASTx method, demonstrated variable performance for DNA virus samples when compared with direct nucleotide classification. Again, regardless of host sequence retention MegaHIT yielded slightly higher sensitivity than metaSPAdes assemblies, but differences were overall minor and more frequent in strain than species matching. For assessing RNA viruses, we first discounted two samples which were spiked with the viroids Peach latent mosaic viroid (PLMVd) and Hop stunt viroid (HSVd) as these entities do not encode protein sequences and therefore could not be detected with protein matching searches. The median sensitivity for nucleic acid databases ranged from 81.6% to 98.9% for strain matches and 91.0% to 99.6% for species-level matches (Figure 4). By comparison, after accounting for the viroid samples, strain-level sensitivity for BLASTx searches was 81.4% for the genome_prot database and 90.9% for nr_derep1 and nr_derep2 for strain matches, and 90.9%, 99.6%, and 98.9% for species matching (Figure 4). We attribute the finding of equivalent species sensitivity but inferior strain sensitivity to the wider range of target sequences reported by the BLASTx search, which likely fulfilled the maximum target sequence requirement of the BLASTx search before the correct strain targets could be encountered.

### 3.7. Performance of Protein–Protein Classification Approaches

Instead of BLASTx searching, an alternative method for acquiring rigorous protein annotations is to perform the translation step independently of the classification process to make more explicit use of the known features of protein-coding regions to refine the query sequences. Initial comparison between prodigal and TransDecoder revealed that a more stable gene prediction outcome across samples was produced by prodigal (Table 6). Protein classification was performed for each set of predicted proteins using usearch and DIAMOND BLASTp against the protein databases (Table 2), and HMM searching was performed against the mcl_4.0 database.

At the species level, protein predictions from DNA and RNA viruses (excluding viroids) were sufficient to produce 100% sensitivity under most assembly and host removal conditions (Figure 5). We observed slightly greater sensitivity when using DIAMOND compared with usearch for DNA virus classification, but results were identical for RNA viruses. Sensitivities were poorer when examining the predictions made by TransDecoder even when accounting for the number of samples from which protein-coding sequences were successfully produced (Appendix A).

HMM searches against the mcl_4.0 database yielded strong detection sensitivity, especially when considering that the protein models were produced from only the genome_prot sequences and lacks most of the sequences in the nr_derep1 and nr_derep2 databases (Table 2). The mcl_4.0 database outperformed all other protein databases at the strain level (Figure 5) and performed markedly better than the genome_prot database, from which it was generated, at the species level (median sensitivity 98.2% versus ~90.5%). At the resolution of species, the nr_derep1 and nr_derep2 databases both outperformed mcl_4.0 by a small margin, with a sensitivity of 99.3%. However, it is worth noting that the mcl_4.0 database alone cannot predict the species of the proteins detected by the models and a secondary BLASTp was required to assign taxonomy to the proteins identified by the HMM approach. Due to the need for this secondary step, we do not believe that the HMM approach would be ideal for routine diagnostic cases, but it’s superior ability to capture viral protein sequences which are not already present in the database, as shown by the improved sensitivity when compared with the genome_prot database (Figure 5), demonstrates the utility of this approach for detection of novel viral sequences.

### 3.8. Performance of Analysis Workflows against VIROMOCK Data

To further validate our results, we obtained 18 datasets from the VIROMOCK challenge (https://gitlab.com/ilvo/VIROMOCKchallenge, accessed 05/08/2021). These data comprise a mixture of real, synthetic, and artificially spiked sequence libraries to provide an independent test panel for the development of bioinformatic pipelines for virus detection. All datasets were downloaded, quality-filtered, and assembled using MegaHIT, SPAdes, and the appropriate viral SPAdes workflow for the viruses in the dataset. This yielded three assemblies for each dataset, except for dataset 4 which contains a mixture of DNA and RNA viruses, so was assembled using MegaHIT, metaSPAdes, rnaSPAdes, and rnaviralSPAdes. Each set of assembled contigs were then classified using all nucleic acid methods, and coding sequence predictions were made with prodigal for protein annotation. Detection of the expected viruses was achieved in most cases for at least one assembly option (Appendix A) although as with the simulated libraries, there was no clearly superior assembly tool, with both MegaHIT and rnaSPAdes achieving greater sensitivity in different datasets.

With the previously noted exception of the protein annotation methods being unable to detect viroid sequences, there were only two instances where a detection could not be made. In the case of dataset 1, we were unable to identify Apple hammerhead viroid (AHVd) sequences in any assembly with any classification tool (Appendix A). However, we note that the documentation for this dataset contains a disclaimer that the presence of this viroid could not be validated by qPCR and therefore may be the result of contamination. Investigation of the unassembled reads revealed just a single read pair which could be classified as AHVd. Due to this finding, and the fact that this dataset is from *Citrus* × *sinensis* and AHVd has an extremely limited host range [109,110], we deem this unlikely to be a true instance of the viroid. For dataset 9 we were unable to produce a strain-level match for the Pistacia emaravirus B, but we did identify sequences classifying to the genus *Emaravirus* within our assemblies. Under standard diagnostic settings such a finding would prompt further investigation, so we considered this a successful detection event although it did not meet the exact requirement of the challenge.

### 3.9. Challenges in Creating a Stand-Alone Analysis Pipeline

Despite its clear value in the field of biosecurity, there are still no internationally standardised pipelines for the detection of viruses using high-throughput sequencing. This challenge stems primarily from the difficulty in determining a ‘best’ tool for each stage of analysis. Variation is one of the foundational concepts of biology and a consequence of genetic variation within and between species means that there is typically no single best tool for a job, but a set of tools which outperform each other under different conditions. We therefore considered not only the ability of our selected tools to achieve accurate results, but also how robust they were to different starting libraries to ensure that our final workflow could be universally applied to diagnostic samples without the need to customise analysis parameters or repeat analysis steps due to computational crashes.

There is also a need to validate HTS-based detections, as false-positive results are possible when such large numbers of sequence data are surveyed, particularly when using reduced-representation classification approaches. Within our workflow there was no need for performing downstream validation as the viral species within each sample was known in advance. In application, however, some form of verification would be required following from the initial detection, either purely bioinformatic (i.e., recovering a complete genome) or using traditional diagnostic assays to provide an independent observation of the virus in question.

## 4. Conclusions

We assessed a total of 60 tool and database combinations, either sourced from the published literature or developed in house, for their efficacy as methods for the detection of plant and animal viruses. Despite the development of a plethora of tools for the express purpose of recovering viral sequences from metagenomic and RNAseq datasets we found that more generic alignment and k-mer based classification approaches provided superior sensitivity. We attempted to improve upon the speed required for classification with these tools by creating more streamlined virus databases for classification. To benchmark the performance of candidate databases and classification approaches we tested nearly 350 mock datasets using a combination of assembly, gene prediction, and classification approaches to establish an end-to-end analysis workflow which is rapid to run and robust across library preparations. Our work provides a strong foundation for the creation of an automated analysis pipeline and will be of value as HTS-based diagnostics become increasingly common in the biosecurity field.

## Figures and Tables

**Figure 1 viruses-14-02163-f001:**
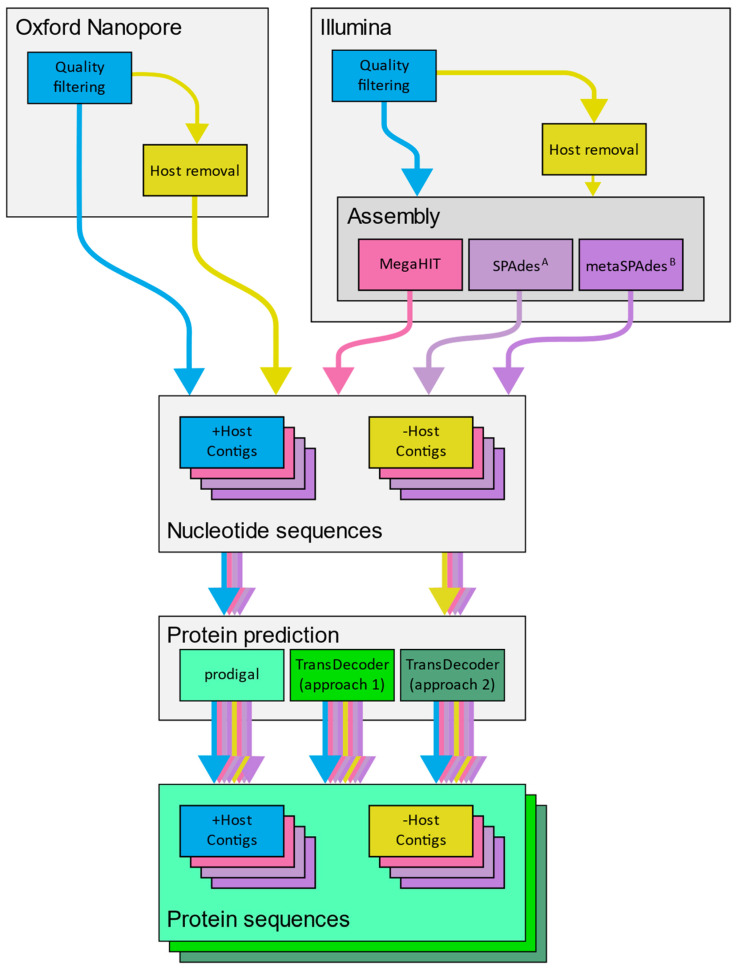
The flow of analysis stages for each simulated library, depending on whether it was produced as an ONT or Illumina dataset. Each set of nucleotide and protein sequences were classified and screened for the virus of interest. Each traversal from raw sequences to a set of nucleotide and protein sequences is referred to as an analysis iteration.

**Figure 2 viruses-14-02163-f002:**
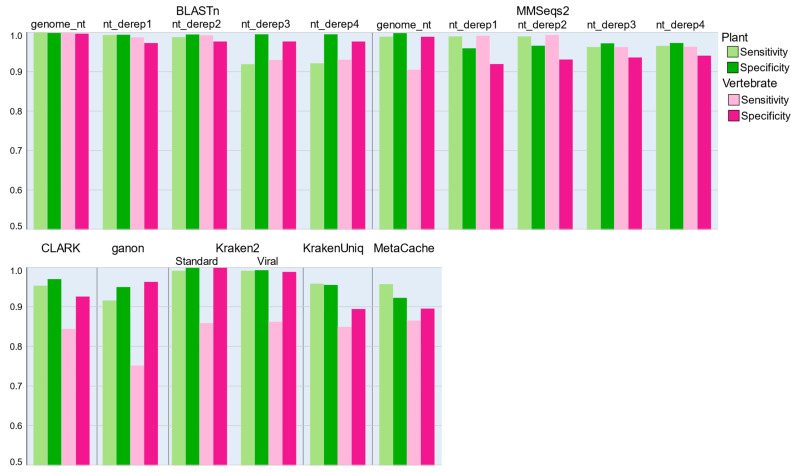
Sensitivity for detection of plant- and vertebrate-infecting viruses, and specificity against plant and vertebrate sequences, using nucleotide classification methods. (Top) BLASTn and MMSeqs2 classifications against each of the in-house nucleotide databases, after filtering for results with at least 70% identity over 50% of the query length and an e-value of less than 0.001. (Bottom) Classification using k-mer profiling tools after result filtering with the appropriate classification threshold for each tool. Classification was performed against a single database, except for Kraken2, in which case both the standard and viral databases were tested.

**Figure 3 viruses-14-02163-f003:**
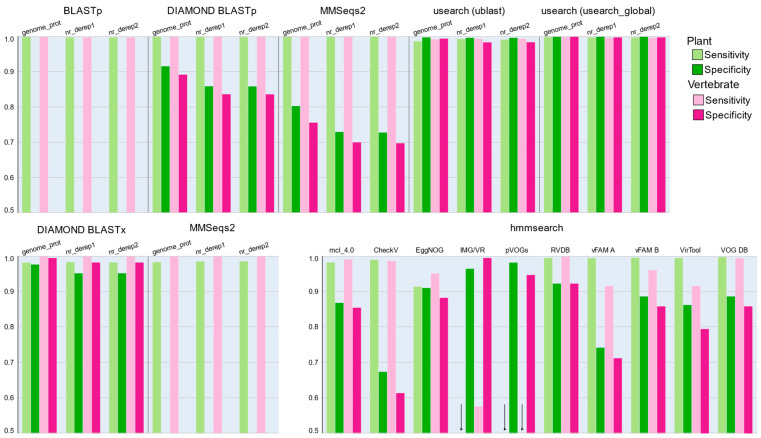
Sensitivity for detection of plant- and vertebrate-infecting viruses, and specificity against the plant and vertebrate sequences, using protein annotation methods. Databases used for classification are provided under each tool name. (Top) BLASTp, DIAMOND BLASTp, MMSeqs2, and usearch (ublast and usearch_global methods), after filtering for results with at least 70% identity over 50% of the query length and an e-value of less than 0.001. (Bottom, left) Nucleotide/protein translated searches using DIAMOND BLASTx and MMSeqs2. (Bottom, right) Classification according to HMM searches against candidate profile databases. For translated searches and HMM profile searches filtering was performed specifying an e-value of less than 0.001. Arrows denote instances where classification was successful, but at a rate too low to be visible in the *y*-axis. Due to impractical run time requirements, BLASTp and translational MMSeqs2 approaches could not be assessed for specificity.

**Figure 4 viruses-14-02163-f004:**
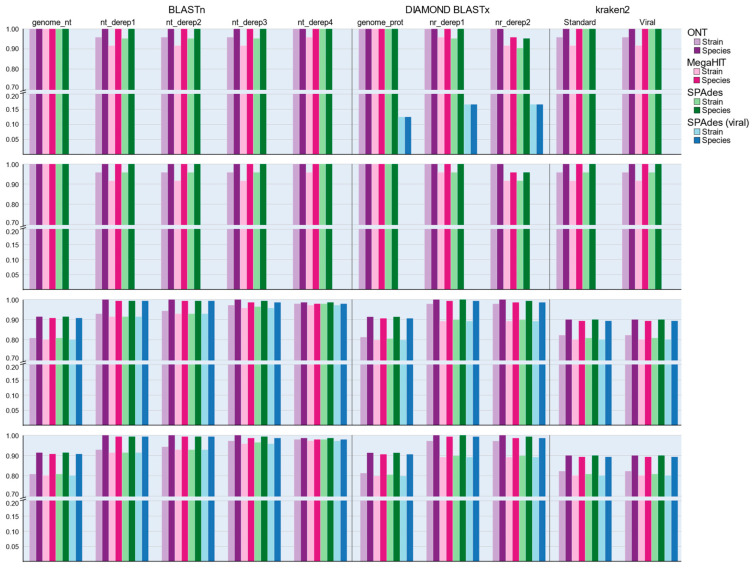
The rate of successful detections at either the strain or species level for each mock library analysis options for nucleic acid classification tool. Sequence sensitivity for each tool and database combination is reported as the proportion of successful identification events across all samples for the given sequencing platform, as assessed at strain- or species-level resolution.

**Figure 5 viruses-14-02163-f005:**
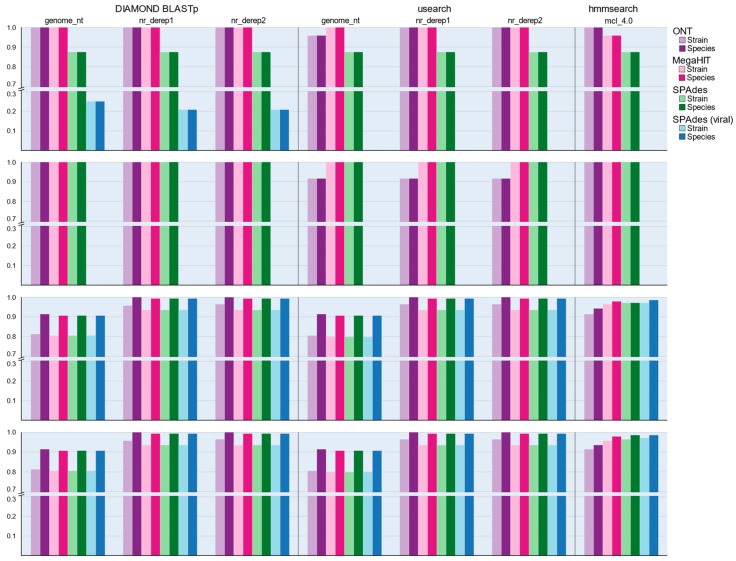
The rate of successful detections at either the strain or species level for each mock library analysis from coding region predictions made using prodigal. Sequence sensitivity for each tool and database combination is reported as the proportion of successful identification events across all samples for the given sequencing platform, as assessed at strain- or species-level resolution.

**Table 1 viruses-14-02163-t001:** A summary of the number of genomes, unique nucleotide contigs, and representative protein coding sequences present in each of the non-target lineages.

Lineage	Genomes	Unique Contigs	Representative Proteins
Fungus	2523	14,840,506	8,485,622
Invertebrate	1137	136,042,698	5,593,319
Plant	769	100,686,597	6,412,958
Protozoa	465	4,690,439	1,701,139
Vertebrate (mammal)	536	198,838,325	3,611,964
Vertebrate (other)	793	52,609,300	5,459,166
Prokaryote	31,910	3,357,372	63,089,651

**Table 2 viruses-14-02163-t002:** Summary statistics for the candidate nucleotide and protein sequence databases constructed for testing.

Sequence Type	Clustering Method	Lineage Groups	Sequences	Database Name
Nucleotide (genome)	Unique contigs	17,231	27,029	genome_nt
Nucleotide (nt)	Species, 90%	40,114	181,612	nt_derep1
	Truncated species, 90%	14,630	157,271	nt_derep2
	Truncated species, 97%	14,630	559,726	nt_derep3
	Truncated species, 99%	14,630	1,076,325	nt_derep4
Protein (genome)	Unique sequences	17,231	686,559	genome_prot
Protein (nr)	Species, 90%	38,457	2,924,049	nr_derep1
	Truncated species, 90%	14,542	2,198,163	nr_derep2

**Table 3 viruses-14-02163-t003:** Number of protein clusters yielded by each method following the removal of duplicate and polyprotein sequences. The final, filtered count of sequences was obtained by removing protein clusters whose membership did not sufficiently cover at least one viral subfamily (see methods for how coverage was assessed). Family and subfamily classifications were exported from the NCBI taxonomy record for each virus species.

Method	Database Name	Initial Clusters	Non-Singleton Clusters	Filtered Clusters
usearch (50%)	usearch_50	253,379	90,165	6612
usearch (60%)	usearch_60	283,612	93,286	6802
usearch (70%)	usearch_70	315,140	94,825	7310
usearch (80%)	usearch_80	350,574	94,751	6514
usearch (90%)	usearch_90	400,303	91,950	5918
mcl (I = 1.4) *	mcl_1.4	135,206	58,272	397 *
mcl (I = 4.0)	mcl_4.0	154,114	77,032	5950
Mixed (usearch 50%)	mixed_50	361,297	110,525	8569
Mixed (usearch 60%)	mixed_60	396,879	119,293	9468
Mixed (usearch 70%)	mixed_70	433,833	126,782	10,283
Mixed (usearch 80%)	mixed_80	475,194	133,293	10,519
Mixed (usearch 90%)	mixed_90	532,477	139,738	9498

Asterisk (*) denotes where clustering was filtered at the family level rather than subfamily.

**Table 4 viruses-14-02163-t004:** Sensitivity, specificity, and average scoring values for the four top performing candidate databases after e-value filtering (0.001) of hits.

Database	Plant Sensitivity	Specificity	F1 Score	Animal Sensitivity	Specificity	F1 Score	Average Sensitivity	Specificity
usearch_50	0.9795	0.8895	0.9323	0.9401	0.8869	0.9127	0.9598	0.8882
mcl_1.4	0.949	0.9285	0.9386	0.8464	0.9022	0.8734	0.8977	0.9153
mcl_4.0	0.9822	0.8679	0.9215	0.9905	0.8541	0.9172	0.9864	0.861
mixed_90	0.9824	0.8468	0.9096	0.9903	0.8152	0.8943	0.9864	0.831

**Table 5 viruses-14-02163-t005:** Detection sensitivity for nucleotide sequences using virus detection pipelines. Note that the sensitivity of VirTool was assessed using only the sequence mapping portion of the workflow—the VirTool HMM database was assessed as part of the HMM testing later in this manuscript.

Tool	Database	Plant	Vertebrate	Invertebrate	Other
DeepVirFinder	-	0.6654	0.6039	0.7223	0.9104
FastViromeExplorer	eukvir	0.0279	0.0102	0.0064	0.0030
	gov	0.0000	0.0000	0.0011	0.0002
	imgvr	0.0000	0.0000	0.0000	0.0000
	ncbi	0.0401	0.0138	0.0070	0.0115
MiCOP	-	0.8875	0.7458	0.8947	0.8073
VirFinder	default	0.4442	0.3654	0.4629	0.8437
	modEPV	0.9149	0.8083	0.8668	0.8539
ViromeScan	-	0.9149	0.8699	0.5994	0.7890
VirSorter	db1	0.0098	0.0978	0.0199	0.7397
	db2	0.0098	0.0943	0.0199	0.7316
VirSorter2	-	0.8664	0.9411	0.9635	0.9843
VirTool	-	0.8855	0.0063	0.0086	0.0093

**Table 6 viruses-14-02163-t006:** The number of successfully completed assemblies for each assembler and subsequent coding region predictions with each approach. The number of samples processed were 24 for DNA and 141 for RNA libraries.

Host	Library Type	Platform	Assembler	Prodigal	TransDecoder	TransDecoder (Guided)
Retained	DNA	ONT	-	24	14	24
		HiSeq	MegaHIT	24	24	24
			SPAdes (meta)	21	21	17
			SPAdes (metaviral)	24	24	24
	RNA	ONT	-	141	141	140
		HiSeq	MegaHIT	141	141	141
			SPAdes (rna)	141	141	141
			SPAdes (rnaviral)	141	141	141
Removed	DNA	ONT	-	24	24	24
		HiSeq	MegaHIT	24	14	14
			SPAdes (meta)	24	17	17
			SPAdes (metaviral)	0	-	-
	RNA	ONT	-	141	113	113
		HiSeq	MegaHIT	141	130	130
			SPAdes (rna)	141	140	140
			SPAdes (rnaviral)	141	133	133

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
