# Peer review of "Development and Validation of a Bioinformatic Workflow for the Rapid Detection of Viruses in Biosecurity"

_viruses, 2022, doi:10.3390/v14102163_

Round 1
Reviewer 1 Report
Authors made a great comparison of performance of various tools used for bioinformatic processing of HTS data, oriented towards a biosecurity measures against plant and animal viruses.
The work was performed at artificial datasets and custom virus databases, all in nucleotide and protein levels.
Sensitivity and specificity of various tools were assessed on artificial databases. Only the best performing tools were selected for the performance tests.
They showed the generic tools like BLAST and its variants to be best perfomers for a such task, showing viral-specific tools to be less efficient.
The cause of the problem seems to be in a large extent of a virus genome structures compared to other organisms. They also suggested that the virus-specific tools were optimized for a phage genomes, which are just a one group in a large and divergent realm of viruses.
The manuscript deserves publication.
Reviewer 2 Report
The manuscript addresses the topic of defining standardized pipelines for virus detection using high-throughput sequencing. The authors generated a high number of simulation data and as well took advantage of available datasets to compare different pipelines for plant and animal virus detection at nt and aa level. The theme is of high interest, giving the increasing implementation of HTS for virus detection. Pipelines standardization is a key theme in this context, and also challenging to address, like demonstrated by the results of the current study. The definition of a standard bioinformatic workflow is indeed difficult to address, also because bioinformatics is extremely dynamic. Overall, the manuscript is strongly focused on bioinformatics, and very dense to read. I do not have specific comments on the content at this stage; a possible consideration would be on the journal to be considered for submission, given the strong bioinformatics focus.